# Vinyl Esters and Vinyl Sulfonates as Green Alternatives to Vinyl Bromide for the Synthesis of Monosubstituted Alkenes via Transition-Metal-Catalyzed Reactions

**Tomáš Tobrman** 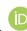

Department of Organic Chemistry, University of Chemistry and Technology, Prague, Technická 5, 16 628 Prague, Czech Republic; tomas.tobrman@vscht.cz

**Abstract:** This review summarizes the applications of vinyl sulfonate and vinyl acetate as green alternatives for vinyl bromide in cross-coupling reactions. In the first part, the preparation of vinyl sulfonates and their cross-coupling reactions are briefly discussed. Then, a brief review of vinyl acetate cross-coupling reactions, including cyclization reactions, the Fujiware–Moritani reaction, and transvinylation reactions are described.

**Keywords:** vinyl acetate; vinyl sulfonate; cross-coupling reaction; transvinylation

## 1. Introduction

Cross-coupling reactions have become an indispensable aspect of modern organic synthesis [1–5]. The applications of cross-coupling reactions involve the formation of C–C and C–heteroatom bonds, including C–B [6–10], C–N [11–15], C–O [16–19], C–P [20–24], and C–S bonds [25–28]. A wide range of compounds with applications in medical or material science has been synthesized by means of cross-coupling reactions. Here, the synthesis of tetrasubstituted alkenes [29–37] and the preparation of heterocyclic compounds [38–43] are examples of such syntheses. A typical reaction scheme for the formation of a C–atom bond via a cross-coupling reaction involves a reaction between an electrophile and a nucleophile. Alternatively, it is also possible to form bonds between two electrophiles or two nucleophiles, with the related reactions being known as oxidative [44–46] or reductive [1,47–49] cross-coupling reactions, respectively.

A traditional cross-coupling reaction requires a mutual interaction between the nucleophile and the electrophilic reagent, which is catalyzed by a transition metal complex (Scheme 1a). Initially, the transition metal complex reacts with the electrophile to produce the product of the oxidative addition. This is followed by the transmetalation step, while in the final step, which is known as reductive elimination, the reaction product is formed alongside the active catalytic species, which represents the beginning of the next catalytic cycle. Among the wide range of available nucleophilic reagents, boronic acids, organostannanes, organozinc compounds, Grignard reagents, alkenes, and alkynes are commonly used for the formation of C–C bonds. In contrast, organohalides are typical examples of electrophiles that have been widely used in cross-coupling reactions during the past century. In addition to substrates with activated C–N [50–53], C–S [54–56], or C–P [22] bonds, substrates with activated C–O [51,57–61] bonds are often used. Together with traditional cross-coupling reactions, C–H bond activation reactions are being intensively developed (Scheme 1b) [62]. A simplified mechanism for cross-coupling reactions with C–H bond activation illustrates the unpretentiousness of C–H bond activation on the structure of the starting materials. However, the mechanisms of C–H bond activation are considerably more complex [63].

**Scheme 1.** A simplified mechanism for a general transition-metal-catalyzed cross-coupling reaction. (**a**) General cross-coupling reaction; (**b**) General cross-coupling reaction including C–H activation.

From a practical perspective, substrates with activated C–O bonds are of particular interest. These substances are represented by various esters, sulfonates, and phosphates. Organophosphates and organosulfonates are often used as electrophiles in cross-coupling reactions due to their easy availability. Indeed, the easy availability of organosulfonates and organophosphates is demonstrated by the preparation of vinyl derivatives **3** and **4** (Scheme 2a). The starting ketone **1** is enolized by means of a base and then the formed enol **2b** is phosphorylated [64] or sulfonylated [65] at the O-terminus. A variation of this approach is used to synthesize aryl sulfonates [66] and aryl phosphates [67]. In addition, the Perkow reaction [68], which is based on the reaction of halogenated ketones with phosphites, can also be used to prepare vinyl phosphates (Scheme 2b). This is particularly advantageous for the preparation of triple [69,70] and double [71–76] electrophilic templates **6** using brominated ketone **5**.

**Scheme 2.** Synthesis of vinyl sulfonates and vinyl phosphates. (**a**) An acid-base reaction for the preparation of vinyl sulfonates and vinyl phosphates; (**b**) The Perkow reaction for the preparation of vinyl phosphates.

Cross-coupling reactions are also useful for the introduction of simple fragments—namely, the vinyl group. Traditionally, dehydrohalogenation [77], the Wittig reaction [78–82], or the Peterson olefination [83,84] were commonly used to introduce the vinyl group, although the discovery of cross-coupling reactions has expanded the available vinylation methods. In principle, vinylation can be performed via the reaction of metalated ethylene with an electrophilic reagent in the presence of a catalytic amount of a transition metal complex (Scheme 3). Vinylboronic acid [85–87], vinylmagnesium bromide [88], and vinylstannanes [89,90] have all been used en route to monosubstituted ethylene. Another procedure—that is, a reaction of organometallic reagents with vinyl-type electrophiles, mainly vinyl bromide—is also widely used. The opposite approach, which is based on the cross-coupling reactions of vinylic electrophiles, is also feasible. The Sonogashira reaction [91–94], radical hydroarylation [95], and the Suzuki cross-coupling reaction [96]

are typical examples of the use of vinyl bromide in organic synthesis. However, vinyl bromide has a low boiling point (15.8 °C) [97], while the International Agency for Research on Cancer has listed vinyl bromide as a suspected human carcinogen [98].

**Scheme 3.** General scheme representing vinylation under transition metal catalysis.

The aforementioned properties of vinyl bromide make it a difficult and dangerous substrate to manipulate. Fortunately, vinyl electrophiles with activated C–O bonds can be considered a suitable alternative to vinyl bromide for use in cross-coupling reactions, especially in the formation of C–C bonds. As no prior review has summarized the use of vinyl esters with activated C–O bonds, the aim of this review is to summarize the preparation and use of the compounds listed in Scheme 4 in terms of the preparation of monosubstituted ethylenes **8** and disubstituted alkenes **9**. Vinyl sulfonates **7a** and vinyl acetates **7b** are easily available and frequently used in organic synthesis, while vinyl phosphates **7c** and vinyl carbamates **7d** are readily available but are characterized by undeveloped chemistry. Vinyl carbamates are frequently used as acetylene surrogates due to their α-lithiation-elimination sequence [99,100], whereas vinyl phosphates **7c** are rarely used in organic synthesis [101]. Therefore, this review focuses on the chemistry of vinyl sulfonates **7a** and vinyl acetates **7b** and covers the 2015–2023 period. Although the C–H bond activation reactions of substrates **7a** and **7b** do not directly provide the vinylation products, selected C–H bond activation reactions for these substrates are covered by this review.

*General reaction scheme*

*Substrate scope*

**Scheme 4.** Scope of this review.

## 2. Cross-Coupling Reactions of Vinyl Sulfonates

Despite the existence of alternative processes, most vinyl sulfonates are currently prepared via the decomposition of tetrahydrofuran with butyllithium at 35 °C. The resulting enolate can be sulfonylated with tosyl chloride [102,103], nonafluorobutanesulfonyl fluoride [104,105], mesyl anhydride [106], or *N*-phenyl-bis(trifluoromethanesulfonimide) [107]. Sulfonates **10**, **12**, and **13** were isolated as a liquid with a significantly higher boiling point than that of vinyl bromide. Only triflate **11** was isolated as a solution in tetrahydrofuran (THF) (Scheme 5).

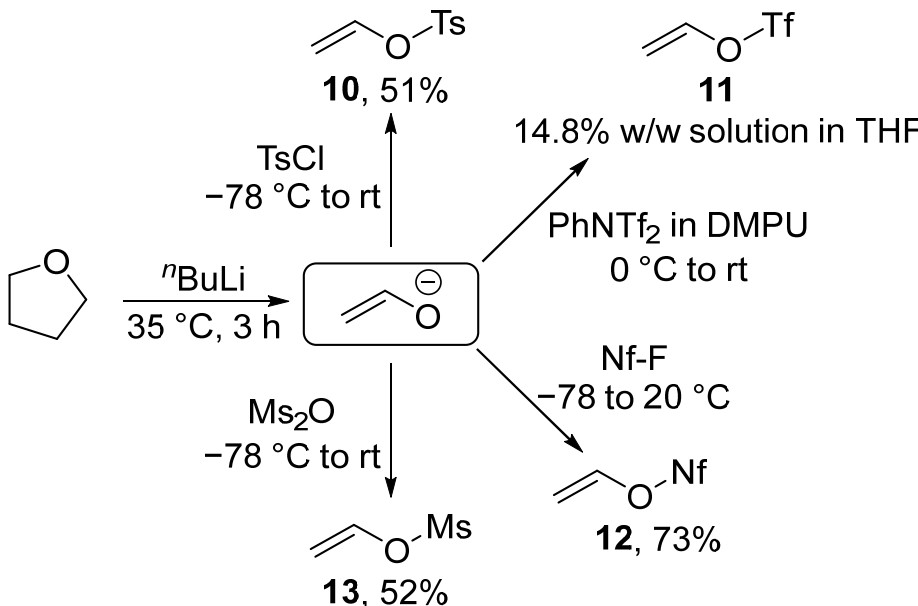

**Scheme 5.** Synthesis of vinyl sulfonates.

The vinylation of alkynylmagnesium bromide **14** can be performed using vinyl nonaflate in the presence of a cobalt catalyst to produce enyne **15** (Scheme 6) [108]. The optimized reaction conditions have a significantly broader scope, although enyne **15** was prepared in a near-quantitative isolated yield. In addition, examples of other selected substrates illustrate how the use of Grignard reagents results in the limited tolerance of the functional groups during this reaction.

$C_8H_{17}$———MgBr   +   ⟋⟍ONf   $\xrightarrow[\text{THF, 2 h}]{\text{Co(acac)}_3 \text{ (3 mol\%)}}$   $C_8H_{17}$———⟍

**14**

**15**, 88%

- - - - - - *Other synthesized enynes* - - - - - -

$C_6H_{13}$———⟍$C_5H_{11}$

**16**, 81%

$C_6H_{13}$———⟍NBoc

**17**, 69%

$Et_3Si$———⟍

**18**, 93%

**Scheme 6.** Cobalt-catalyzed vinylation of a Grignard reagent.

Styrene derivative **20** was obtained via the vinylation of arylmagnesium bromide **19** in the presence of a catalytic amount of palladium complex (Scheme 7) [109]. The same catalyst was also used for the asymmetric methoxycarbonylation of styrene **20** to provide the prodrug of flurbiprofen **21b**. The disadvantage of this procedure for the preparation of the prodrug of flurbiprofen is the formation of an equimolar mixture of linear **21l** and branched product **21b**.

**Scheme 7.** Preparation of the prodrug of flurbiprofen **21b** via the vinylation of Grignard reagent **19**.

Significantly better tolerance of the functional groups is achieved during the cross-coupling reaction of vinyl nonaflate with the organocopper reagent when catalyzed by an iron complex (Scheme 8) [110]. For the preparation of functionalized Grignard reagents, it is necessary to use the approach developed by Professor Knochel [111–114]. The scope of the published cross-coupling reaction is remarkably broad, although only nitrile and ester groups have been used to date.

**Scheme 8.** Iron-catalyzed reaction of vinyl nonaflate with an organocopper reagent.

The Heck reaction is another popular alternative for the preparation of alkenes. For instance, the Heck reaction of vinyl nonaflate with methyl acrylate and styrene led to alkenes **23** and **25** (Scheme 9), as reported in 2001 [105]. In the case of alkene **23**, the vinylation was further extended to prepare conjugated triene **24** under similar reaction conditions and with a good overall yield. Surprisingly, styrene provided substituted buta-1,3-diene in a low isolated yield. This was an unexpected finding because styrene reacts very well under Heck reaction conditions with heteroaryl [115], aryl [116], and vinyl halides [117]. The observed reactivity of styrene was explained by the competitive elimination of NfOH under the utilized reaction conditions.

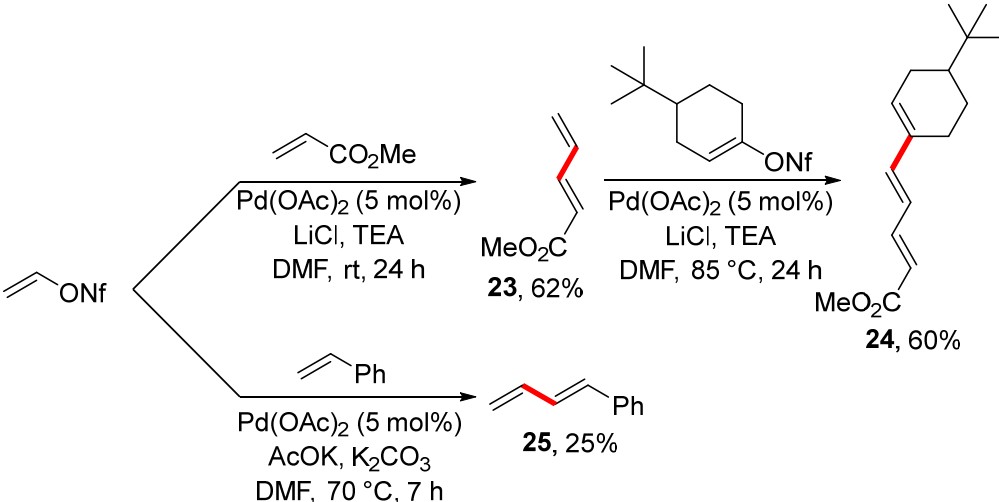

**Scheme 9.** Heck reaction of vinyl nonaflate.

An extensive study concerning the Heck reaction was published by the Skrydstrup group [118]. The Heck reaction of vinyl tosylate gave the expected diene **26**. Cyclic tosylates react similarly, as illustrated by examples **27** and **28** (Scheme 10a). However, the Heck reaction of tosylate with the bulky *tert*-butyl group **30** provided the isomerized product **31** (Scheme 10b). Study of the reaction mechanism by means of quantum chemical calculations showed that the isomerization begins with the oxidative addition of the palladium complex to form the T-shaped complex **A**. Subsequently, the β-elimination of the H–Pd moiety forms complex **B**. Then, the isomerization to complex **C** occurs through reinsertion of *tert*-butylacetylene with the opposite regioselectivity. Complex **C** further undergoes the Heck reaction.

Vinyl tosylate has been successfully used in C–H activation reactions. The palladium-catalyzed vinylation of benzoxazole **32** was described by Lei and Kwong (Scheme 11) [119]. The product of vinylation reaction **33** was isolated in a 70% yield. The scope of the described reaction is limited to simple vinyl tosylates, as illustrated by the other used tosylates. In addition, selected synthesized 2-substituted benzoxazoles were used for the preparation of cyclometalated Ir(III) complexes.

*(a)*

**Scheme 10.** Study of the Heck reactions of various vinyl tosylates. (**a**) The Heck reaction of vinyl tosylate; (**b**) 1,2-Migration during the Heck reaction.

**Scheme 11.** Palladium-catalyzed C–H activation for the vinylation of benzoxazole.

Yu developed reaction conditions for the photoredox nickel-catalyzed vinylation and arylation of anilines (Scheme 12) [120]. Here, *N,N*-dimethylaniline (**34**) reacted with vinyl tosylate in the presence of ruthenium and nickel complexes to yield *N*-allylaniline **35**. This reaction is characterized by a wide range with respect to vinyl sulfonates, including triflates derived from lactone **36** and lactame **37**. The detailed mechanism of the transformation was not discussed; however, the authors showed that the procedure can be used, for example, for the modification of estrone **39** via its conversion into the corresponding triflate followed by photoredox nickel-catalyzed C–H arylation.

**Scheme 12.** Photoredox/nickel dual catalysis en route to C–H vinylation.

Multicomponent reactions represent another class of transformation of vinyl sulfonates. Examples of multicomponent reactions are presented in Schemes 13 and 14. The first multicomponent reaction of vinyl sulfonates involves the reaction of vinyl nonaflate, phenylboronic acid, and alkene in the presence of a catalytic amount of palladium complex to give alkene **41** in a quantitative yield (Scheme 13) [104]. The proposed mechanism begins with the formation of complex **A** via the oxidative addition of the palladium(0) complex followed by migratory insertion. Then, β-hydride elimination forms complex **B**, which is hydropalladated to provide the η³-allyl complex **C**. Subsequently, transmetalation followed by reductive elimination gives the product of reaction **41**.

**Scheme 13.** The use of vinyl nonaflate in a multicomponent reaction.

**Scheme 14.** Carbopalladation of a C–C bond en route to vinylated cyclobutene.

Aggarwal et al. discovered a new multicomponent reaction of vinyl triflate, which involves the C–C bond carbopalladation of strained bicyclo[1.1.0]butyl boronate (Scheme 14) [107]. A selected example of vinylation involves the reaction of vinyl triflate with the ate complex **44** generated from sulfon **44**, *tert*-butyllithium, and cyclohexan-1-ylboronic acid pinacol ester **43**. The reaction mechanism entails the oxidative addition of the Pd(0) complex to vinyl triflate, which is followed by carbopalladation by means of the cationic Pd complex **B**. The final product **46** is formed via reductive elimination. Although the presented example is limited to vinyl triflate, the paper focused on aryl triflates and high diastereoselectivity (d.r. > 98:2) was achieved in all cases.

### 3. Cross-Coupling Reactions of Vinyl Acetates

Vinyl acetate is a popular reagent for the vinylation of various substrates. The reason for its popularity is its commercial availability and ease of preparation from acetic acid and ethylene under transition metal catalysis, including the use of Pd–Au catalysts (Scheme 15) [121].

**Scheme 15.** Synthesis of vinyl acetate.

The Kumada reaction of vinyl acetate, as an example of traditional cross-coupling chemistry, was extensively studied by Wangelin (Scheme 16) [122]. The reaction was catalyzed by ferric chloride and the optimized reaction conditions were used to prepare a series of di- and trisubstituted alkenes, including monosubstituted ethylene derivatives **47**. The authors obtained valuable data through mechanistic studies. The study of the reaction mechanism revealed the following patterns: (i) kinetic poisoning studies showed that the studied reaction is a homogeneously catalyzed reaction; (ii) the coordination of the alkene is the rate determining step; and (iii) vinyl acetates are more reactive than vinyl pivalates and vinyl carbamates.

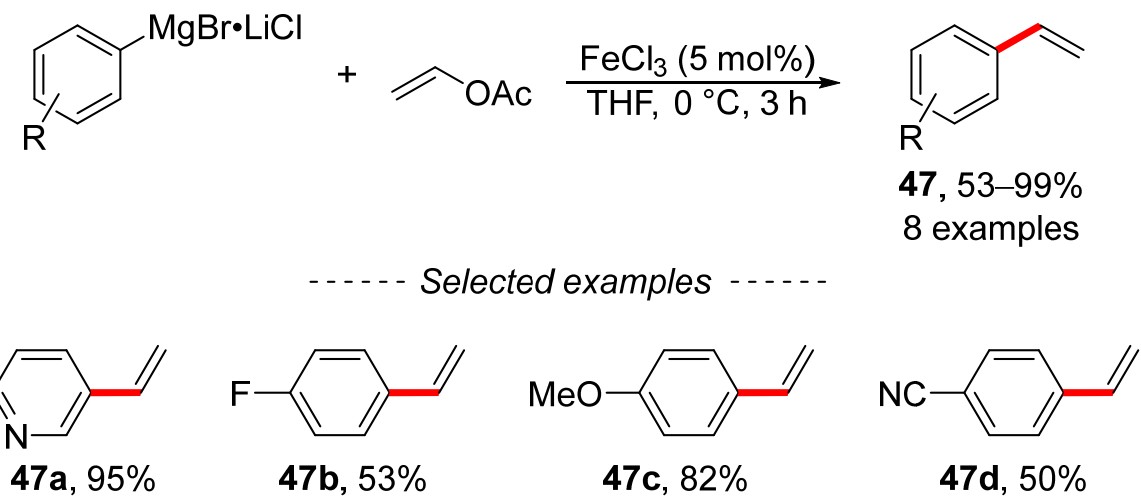

**Scheme 16.** Cross-coupling reaction of vinyl acetate with arylmagnesium bromides.

In addition to the Kumada reaction, vinyl acetate can also be used in reductive cross-coupling reactions. A recent example describes the vinylation of organobromide **48** when catalyzed by a nickel complex (Scheme 17) [123]. The described vinylation is characterized

by simple experimental conditions with a wide scope. Moreover, 88 compounds were prepared, including vinylated aromatic and heteroaromatic hydrocarbons. The proposed mechanism is based on standard knowledge in the field of reductive cross-coupling reactions.

**Scheme 17.** Nickel-catalyzed reductive cross-coupling of vinyl acetate with aryl bromides.

The activation of the C–H bond is a popular method for the introduction of a vinyl group by means of vinyl acetate together with rhodium, iridium cobalt, or palladium complexes. The published iridium-catalyzed reaction describes the vinylation of enamides **50** (Scheme 18) [124]. The reaction conditions tolerate a variety of functional groups, including nitro, ester, halogen, nitrile, and Bpin groups. The scope of the reaction is not limited to the preparation of cyclic dienes **51a**, as the acyclic compounds **51b** and **51c** are also available via this method. Based on quantum chemical calculations and experimental results, a reasonable mechanism was proposed. Initially, the activation of the C–H bond

gives complex **A**. Subsequently, the migratory insertion of vinyl acetate into the Ir–C bond is followed by β-elimination to form the reaction product **51**. The authors also showed that the reaction can be used for the preparation of cholestan-3-one derivatives. First, cholestan-3- was converted into the conjugated diene **51d** under the developed reaction conditions. Then, the exocyclic vinyl group was modified by means of hydroboration to give borane **52**. Diene **51d** can also be hydrolyzed to unsaturated ketone **53,** with its oxidation affording the cholestane derivative **54** in only a 21% isolated yield.

**Scheme 18.** Iridium-catalyzed C–H vinylation of enamides.

The C–H vinylation of aromatic ketones can also be catalyzed by a cobalt complex under ligandless conditions (Scheme 19) [125]. The optimized reaction conditions were mostly used for the modification of substituted acetophenones **55**, although the successful vinylation of carbazole **56c** showed that heteroaromatic ketones can also be used. Modification of the vinyl group afforded aldehyde **57**, which completed the formal synthesis of bruguierol A [126]. The results of mechanistic studies are in agreement with the findings reported in the literature. Thus, the presence of silver salt and copper acetate is essential for the formation of catalyst **A** from the starting complex [Cp*CoI₂]₂. Activation of the C–H bond followed by migratory insertion, β-hydride elimination, and reinsertion gives complex **C**. Subsequent β-acetate elimination provides the main reaction product **56**.

**Scheme 19.** Cobalt-catalyzed C–H vinylation of acetophenones.

Ellman [127] and Wei [128] used vinyl acetate for the chemoselective vinylation of aromatic substrates under the catalysis of the same rhodium complex (Scheme 20). In the first case, substituted benzenes **58** with an amide or 2-pyridyl directing group were vinylated. Unfortunately, the scope of this reaction is limited to a few functional groups, which are shown as selected examples **59a**, **59b**, and **59c**. However, the authors were able to demonstrate that rhodacycle **A** is a catalytic resting-state species and that it catalyzes the described reaction. In a similar work, Wei et al. showed that the same rhodium complex catalyzed the vinylation of aromatic hydrocarbons with a sulfonyl ketimine-directing group **60**. The optimized reaction conditions are significantly more complex than in previous

work [127]. In addition, the developed reaction conditions can be used for the vinylation of heteroaromatic compounds **61b** and benzamides **61c**. The applications of synthesized styrenes include the reduction of the imino group of styrene **61d** followed by acid-mediated cyclization into the fused heterocyclic compound **62**.

**Scheme 20.** Selective rhodium-catalyzed vinylation of aromates by means of vinyl acetates.

Vinyl acetate was used for the efficient vinylation of iodouridine **63**, giving vinylated analogue **64** in a 54% isolated yield (Scheme 21a). The isolated alkene **64** was further modified into a new isoindoline-derived benzimidazole nitroxide spin label [129]. An interesting dichotomy was observed during the Heck coupling of chlorinated enamine **65**. The use of allyl acetate gave the Fujiwara–Moritani reaction product **66**, while the use of vinyl acetate gave the product of β-OAc elimination **67** (Scheme 21b) [130].

Interesting reactivity was observed during the palladium-catalyzed modification of 1,2,3-triazole *N*-oxide **68** in the presence of silver carbonate as an oxidant (Scheme 22) [131]. The starting triazole **68** reacted with styrene or methyl acrylate, as well as with other acrylic acid esters, to produce the Fujiwara–Moritani reaction product **69**. However, the use of an electron-rich alkene—namely, vinyl acetate—gave a mixture of the Fujiwara–Moritani reaction product **71** and the vinylation product **70**. The formation of both regioisomers was explained by the activation of the C–H bond by means of palladium acetate. Subsequent migratory insertion of the formed complex **A** and vinyl acetate gave regioisomers **B** and **C**. Then, products **70** and **71** were obtained via β-elimination. It is important to note that the same group described the formation of 4-vinyl-3-(p-tolyl)sydnone during an analogous palladium-catalyzed reaction of vinyl acetate with 3-(p-tolyl)sydnone [132].

**Scheme 21.** Palladium-catalyzed vinylation of organohalides. (**a**) Palladium-catalyzed vinylation of iodouridine; (**b**) The Heck reaction of chlorinated enamine.

An analogous competition between vinylation and Fujiwara–Moritani alkenylation was also observed during the rhodium-catalyzed reaction of vinyl acetate with 7-azaindoles **72** (Scheme 23) [133]. Optimization of the reaction conditions gave regioisomers **73** and **74** in all cases. The optimized reaction conditions were also successfully used for substituted 7-azaindoles, vinyl benzoate, vinyl pivalate, vinyl butanoate, and substituted vinyl acetate. In total, 13 substituted 7-azaindoles were prepared as a mixture of isomeric products **73** and **74,** where the latter was the major product. The disubstituted alkenes **74** were always formed as a mixture of the two stereoisomers, where the *E* isomer dominates. The proposed mechanism for the formation of both products involves several important steps. Initially, complex **A** is formed via the activation of the C–H bond. Then, the coordination of vinyl acetate to complex **A** and the subsequent migratory insertion gives complex **B**, which isomerizes into complex **C**. Both stereoisomers are formed by means of β-H elimination or β-OAc elimination.

Wen and Zhang succeeded in the selective C–H olefination of electron-rich alkenes, including vinyl acetate, when catalyzed by rhodium complexes (Scheme 24a) [134]. An amide group serves as the main directing group, as illustrated by selected examples **75a** and **75b** and the sophisticated Weinreb amide group **75c**. By contrast, acetamido group **75d** can also be used to facilitate C–H activation. A series of mechanistic studies was also performed in this work, the results of which essentially support previous results [133]. In further work, the same authors again used the rhodium complex for the bisindolylation of vinyl acetate (Scheme 24b) [135]. This later work is characterized by the extensive scope and good tolerance of the sensitive functional groups, including the aldehyde, ester, and nitrile groups. Moreover, 2-pyridyl is used as the directing group. The proposed catalytic cycle assumes the formation of vinylated indole **78** as the key intermediate, which is further converted into the product. In a subsequent publication, the authors showed that the

rhodium complex [Cp*RhCl$_2$]$_2$ in combination with AgSbF$_6$, Cu(OAc)$_2$, and vinyl acetate is a suitable system for Heck-type coupling, leading to substituted vinyl acetates [136].

**Scheme 22.** Vinylation versus Heck-type coupling during the C–H activation of 1,2,3-triazole *N*-oxide.

**Scheme 23.** Rhodium-catalyzed vinylation of carboxamides.

**Scheme 24.** Rhodium-catalyzed Fujiwara–Moritani coupling. (**a**) Rhodium-catalyzed C–H olefination of aromates; (**b**) Rhodium-catalyzed C–H olefination of indole.

Recently, unusual cases of the Fujiwara–Moritani reaction catalyzed by palladium acetate have been described (Scheme 25). For instance, the reaction shown in Scheme 25a is characterized by the unusual regioselectivity of the C–H bond activation, which takes place in the para position [137]. The reaction conditions tolerate a wide range of olefins, including vinyl acetate. However, the anilines used are limited to *N,N*-dimethyl- and *N,N*-diethylaniline. The proposed mechanism was supported by density functional theory (DFT) calculations, and the authors suggested that the para selectivity of the C–H bond activation stems from the presence of acetic acid, which protonates the nitrogen atom. Additionally, 1,2-diMe-*o*-carborane **81** can be modified by means of Fujiwara–Moritani coupling through the palladium-catalyzed activation of the B–H bond (Scheme 25b) [138].

**Scheme 25.** Examples of the palladium-catalyzed Fujiwara–Moritani reaction. (**a**) Palladium-catalyzed C–H olefination of *N,N*-diethylaniline; (**b**) Palladium-catalyzed B–H modification of *o*-carborane.

A different procedure for the modification of vinyl acetate is based on an oxidative Heck reaction using boronic acids. This process has been extensively studied in the past [139]. A recent report that used an oxidative Heck reaction between boronic acids and vinyl acetate was published by Lei (Scheme 26) [140]. This reaction is catalyzed by palladium acetate in the presence of benzoquinone, which is used as a reoxidant. The proposed catalytic cycle involves the formation of the key intermediates **A**, **B**, and **C**. In addition, the beginning of the catalyzed reaction is accelerated by the presence of an acidic environment, which renders the palladium catalyst more electron-deficient. An interesting extension of the oxidative Heck reaction between vinyl acetate and boronic acids entails its implementation in a continuous flow reactor [141,142].

Vinyl acetates represent the substrate of choice for use in cyclization reactions. In this case, the activation of the C–H bond and the suitable structure of the starting material comprise the method of choice. A report published in 2015 described the synthesis of isoquinolines via rhodium-catalyzed cyclization between acetophenone acetyl oximes and vinyl acetates or 1-substituted vinyl acetates. At the end of an extensive study, the cyclization reaction was used to prepare papaverine **88** (Scheme 27) [143]. The synthesis of the target compound **88** starts with carboxylic acid **85**, which is converted into ketone **86** by Friedel–Craft acylation of 1,2-dimethoxybenzene. Acetyl oxime **87** is then obtained via a standard procedure in a 78% yield. The final cyclization makes use of the optimized conditions and vinyl acetate to give papaverine in a 52% yield. Subsequently, Marsden used similar reaction conditions to prepare isoquinolines by means of vinyl acetate, rhodium catalyst, and acetophenone acetyl oxime [144].

**Scheme 26.** Oxidative Heck reaction for the synthesis of vinyl acetates.

**Scheme 27.** Total synthesis of papaverine via the rhodium-catalyzed cyclization of acetyl oxime **87**.

Two complementary procedures for the annulation of benzamides have been reported independently by Jeganmohan [145] and Marsden [146] (Scheme 28). Jeganmohan used *N*-chlorobenzamide **90** together with a cobalt complex and vinyl acetate to prepare isoquinolones **92**. Alternatively, methyl vinyl ketone can be used rather than vinyl acetate to give substituted isoquinolones **91**. After a series of experiments, a reasonable mechanism was proposed. Here, the reaction starts with the formation of catalytic species **A** by means of silver acetate. The subsequent *ortho*-cobaltation and insertion of vinyl acetate yield complex **C**. Then, reductive elimination and re-oxidative addition yield complex **D**. Protonation with acetic acid releases catalyst **A** together with intermediate **E**, which is transferred into reaction product **92** via basic isomerization. Similar steps can be used to explain the formation of substituted isoquinolones **91**. By contrast, Marsden took advantage of rhodium-catalyzed annulation to accomplish the synthesis of similar isoquinolones **93** from the corresponding pivalates **89**. A total of 20 examples of isoquinolones **93** was prepared under the optimized reaction conditions, including protected isoquinolone **93a**. In addition, fused heterocycles were obtained from thiofen-2-carboxylic acid amide **93b** or enamide **93c**. A mechanism for the described conversion has not yet been proposed. However, the intermediate **93d** used for the synthesis of the hepatitis C protease inhibitor MK-1220 was prepared as a single regioisomer in an 85% yield from amide **89a**.

Moreover, enaminones can be used for the synthesis of mainly push–pull substituted naphthalenes via rhodium-catalyzed [5 + 1] annulation of enaminones **94** (Scheme 29) [147]. During the reaction, naphthalene derivatives with formyl and *N,N*-dimethylamino groups are formed. These groups can be further modified by, for example, the addition of a Grignard reagent, Wittig olefination, or oxidation into a nitroso compound. The proposed mechanism for this transformation is shown in Scheme 30. The authors assumed that the catalytic particle **A** is formed by means of potassium acetate. Subsequently, a series of typical steps (C–H activation, migratory insertion, and β–H elimination) produces the intermediate **B**. Further activation of the C–H bond yields the intermediate **C**. Migratory insertion then produces compound **D**, which β-eliminates dimethyl amine and ketone **E**. Overall, the catalytic cycle is completed by the condensation of dimethylamine and ketone **E** and the subsequent hydrolysis of iminium salt **F**.

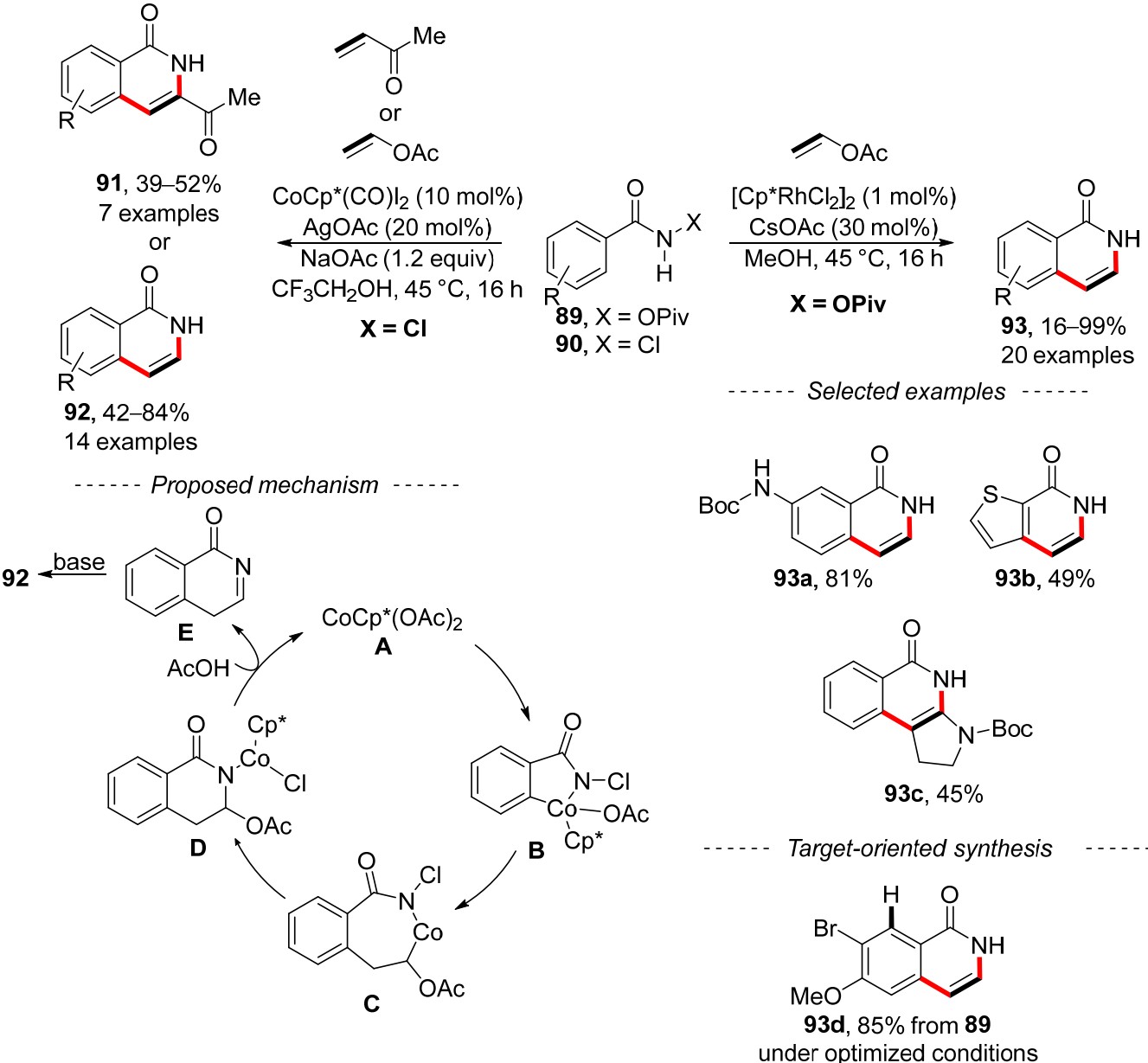

**Scheme 28.** Complementary C–H activation/annulation of benzamides.

In addition to enaminones and acyl oxime (Schemes 27 and 29), the C–H activation/annulation sequence was applied to the synthesis of both isocoumarin **103** [148] and 2-arylquinazolin-4-ones **101** [149] from the corresponding benzoic acid (**100**) or carboxamides **99** (Scheme 31). Moreover, 2-arylquinazolin-4-ones **101** were obtained as a mixture with isomeric product **102** in high overall isolated yields and with limited functional group tolerance (CF₃, F, OMe, CH₃). By contrast, isocoumarin **103** was obtained in a low isolated yield from vinyl acetate. Substantially better isolated yields were obtained for 1-substituted vinyl acetates. Both reactions were catalyzed by the rhodium complex [RhCp*Cl₂]₂, meaning that the proposed mechanisms for the preparation of the target compounds are similar to previous proposals. Additionally, 1-substituted vinyl esters were used for the rhodium-catalyzed transformation of 7-azaindoles into π-conjugated 7-azaindole [150].

**Scheme 29.** Rhodium-catalyzed functionalization of enaminones en route to 1,4-disubstituted naphthalenes.

**Scheme 30.** Proposed mechanism for the formation of 1,4-disubstituted naphthalenes.

**Scheme 31.** Ruthenium-catalyzed C–H activation/annulation transformation of carboxylic acids and carboxamides.

Aside from the above-mentioned C–C bond-forming cross-coupling reactions, vinyl acetate can be used in transvinylation reactions. Transvinylation is typically a transition-metal-catalyzed reaction during which a vinyl group is transferred from vinyl acetate to heteronucleophiles such as carboxylic acids, alcohols, phenols, or amines (Scheme 32).

**Scheme 32.** General scheme representing transition-metal-catalyzed transvinylation.

A frequently used transvinylation reaction entails the preparation of vinyl esters of carboxylic acids. During this reaction, vinyl acetate reacts with carboxylic acid in the presence of a transition metal complex. Palladium acetate is the catalyst of choice for this transvinylation reaction, together with catalytic amounts of protonic acid or potassium hydroxide, as seen in selected examples **104** [151] and **105** [152] (Scheme 33). The reaction is most commonly performed in tetrahydrofuran or with vinyl acetate used as a solvent. In addition to examples **104** and **105**, analogous reaction conditions were used to prepare a wide variety of vinyl esters of carboxylic acids [152–162], including the synthesis of vinyl cinnamates via the flow process [163] and the preparation of vinyl L-leucinate and vinyl L-valinate in the presence of benzoquinone [153]. Transvinylation can also be catalyzed by an iridium complex. This process has recently been used for the preparation of vinyl oleate **106** and vinyl linoleate **107** under harsher reaction conditions when compared with palladium-catalyzed reactions [164,165]. Furthermore, transvinylation between vinyl acetate and adipic [166] or azalaic [167] acid when catalyzed by mercuric acetate and copper acetate has recently been described. This process has only limited use due to the high toxicity of mercuric acetate. In addition, mercury-catalyzed transvinylation proceeds by a different mechanism involving the formation of an acetylene–mercury complex [168].

*Selected examples of recently synthesized vinyl esters of carboxylic acids*

**104** 70%
*Conditions:*
Pd(OAc)$_2$ (10 mol%)
H$_2$SO$_4$ (10 % W/W)
THF, 60 °C, overnight

**105** 15%
*Conditions:*
Pd(OAc)$_2$ (1 mol%)
KOH (10 mol %)
rt, 48 h

**106** 90%

**107** 50%

*Conditions:*
[Ir(cod)Cl]$_2$ (1 mol%)
AcONa (3 mol %)
100 °C, 20 h

**Scheme 33.** Selected applications of transvinylation for the synthesis vinyl esters.

Although the transvinylation reaction has been known for a long time, papers continue to be published that study its course. In 2018, Kadidae et al. described transvinylation as a method suitable for the preparation of vinyl cinnamates **109** (Scheme 34) [157]. The reaction is catalyzed by palladium acetate in the presence of catalytic amounts of potassium hydroxide or sulfuric acid. The choice between basic and acidic catalysis is influenced by the structure of the substrate. Carboxylic acids **109b** and **109d**, with an acid-sensitive functional group, can be vinylated in the presence of potassium hydroxide (and vice versa).

A study published in 2018 showed that transvinylation can be catalyzed by rhodium chloride trihydrate (Scheme 35) [169]. Here, excellent transvinylation results were obtained for simple benzoic acid derivatives. Moreover, excellent chemoselectivity was observed for hydroxybenzoic acids **110a** and **110b**, where exclusive formation of vinyl esters **111a** and **111b** was detected.

**108**

Pd(OAc)$_2$ (10 mol%)
KOH (10 mol%)
or H$_2$SO$_4$ (cat, 10 % W/W)
THF, 40 °C

**109**, 9 examples
16–84%

- - - - - - *Examples of synthesized esters* - - - - - -

**109a**, H$_2$SO$_4$, 21%

**109b**, KOH, 70%

**109c**, H$_2$SO$_4$ 84%

**109d**, KOH, 84%

**Scheme 34.** Recent study concerning the transvinylation of hydroxycinnamic acids.

**Scheme 35.** Rhodium-catalyzed transvinylation.

The mechanism behind palladium-catalyzed transvinylation was studied in the second half of the last century [170,171]. Based on the findings of these studies, it was proposed that the mechanism consists of an oxypalladation–deoxypalladation sequence (Scheme 36). In the first step, vinyl acetate coordinates with palladium acetate to form complex **A**. Then, the activated double bond is attacked by carboxylic acid or carboxylate anion, which leads to complex **B**. Oxypalladation is considered to be the rate-determining step of transvinylation [170]. The resulting complex **B** undergoes deoxypalladation to form both palladium acetate and the reaction product. Transvinylation is also significantly retarded by the substitution of the vinyl group of vinyl acetate [171]. With some simplification, ruthenium- [172] and rhodium-catalyzed [169] transvinylation proceed via analogous mechanisms.

Another popular use of vinyl esters, especially vinyl acetates, entails the preparation of vinyl ethers by means of transvinylation. An earlier review demonstrated that transvinylation can be catalyzed by, for example, palladium, gold, and iridium complexes [173]. However, recent examples **112a** [174] and **112b** [175] illustrate how iridium has become a popular catalyst for use in the preparation of vinyl ethers via transvinylation reactions (Scheme 37).

The main advantage of iridium-catalyzed transvinylation for the preparation of vinyl ethers is the use of a low catalytic amount of the iridium complex and simple reaction conditions. Moreover, Flores-Pérez demonstrated that the iridium complex provided significantly higher yields of vinyl ethers **114** and **116** when compared with mercuric acetate-catalyzed transvinylation (Scheme 38) [176]. In addition, cationic Ir(cod)$_2$BF$_4$ was used for the synthesis of biobased vinyl ethers [177].

**Scheme 36.** Oxypalladation–deoxypalladation sequence en route to transvinylation products.

**Scheme 37.** Selected examples of transvinylation for the synthesis of vinyl ethers.

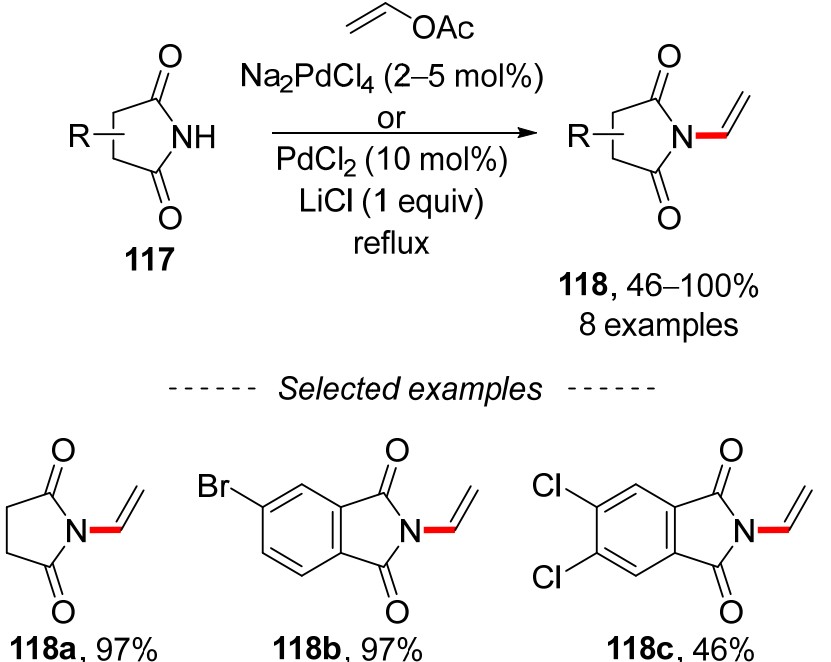

**Scheme 38.** Iridium- versus mercury-catalyzed transvinylation for ether synthesis.

The synthesis of enamines and vinyl amides is another area in which vinyl acetate can be employed. A well-established procedure based on the copper-catalyzed reaction of organohalogens with carboxylic acid amides is termed the Goldberg reaction [178,179]. Aside from organohalogenides, vinyl acetates can also be used. Furthermore, a palladium-catalyzed transvinylation reaction for the preparation of enamides was recently described by Waser et al. (Scheme 39) [180]. This reaction is performed in refluxing vinyl acetate and the isolated yields of vinyl amides **118** are high.

**Scheme 39.** Palladium-catalyzed transvinylation for the synthesis of enamides.

The analogous vinylation reaction can also be used to vinylate *N*-heterocyclic compounds, including carbazole **119** (Scheme 40) [181]. The extension of the optimized conditions to other pyrrole derivatives is limited by the substitution of all positions of the pyrrole ring. Thus, 2,5-dimethylpyrrole did not react under the optimized conditions and the indole derivative was isolated in a 60% yield.

**Scheme 40.** Iridium-catalyzed vinylation of *N*-heterocycles with vinyl acetate.

## 4. Conclusions

This review summarized the available processes for the preparation of mono- and disubstituted ethene by means of electrophilic vinyl templates with activated C–O bonds. Currently, except for low-boiling-point vinyl bromide, vinyl sulfonates and vinyl acetate are most commonly used for simple vinylation. Thus, vinyl sulfonates and vinyl acetates are used to form a single C–C bond via transition-metal-catalyzed reactions. For vinyl acetates, the Fujiwara–Moritani reaction is widely used to prepare monosubstituted vinyl acetates. Due to their easy availability, vinyl acetates are also frequently used for the preparation of vinyl esters and ethers via transvinylation reactions. The reactivity of other electrophiles, particularly vinyl phosphates, is undeveloped, which severely limits the development of multicomponent and cyclization reactions using vinyl templates as a C2 synthon source.

**Funding:** This research received no external funding.

**Data Availability Statement:** Data sharing not applicable.

**Conflicts of Interest:** The author declare no conflict of interest.

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
