# Peer review of "Vinyl Esters and Vinyl Sulfonates as Green Alternatives to Vinyl Bromide for the Synthesis of Monosubstituted Alkenes via Transition-Metal-Catalyzed Reactions"

_chemistry, doi:10.3390/chemistry5040153_

Round 1
Reviewer 1 Report
The review by Tobrman on " Vinyl Esters and Vinyl Sulfonates as Green Alternative to Vinyl Bromide for the Synthesis of Monosubstituted Alkenes via Transition-Metal-Catalyzed Reactions" is a thoroughly detailed and well-structured overview. It effectively addresses the essential and pertinent aspects of this subject matter while also offering valuable insights into potential future directions. The authors have thoughtfully organized the review to summarize the various applications of vinyl sulfonate and vinyl acetate as environmentally friendly substitutes for vinyl bromide in cross-coupling reactions. This manuscript will prove beneficial for readers seeking to gain a comprehensive understanding of the current trends within this field.
However, in some cases, such as scheme 9 and 11, it would be helpful to include a further explanation of the reactions or more detailed chemical mechanisms that help with the understanding of the reaction, because if you are not familiar with this chemistry it could be confused. Here I´ve mentioned some of the points it would be great to clarify, extend the explanation or some detected mistakes. In conclusion, I recommend the acceptance of this review for publication in the Chemistry after minor revision:
(1). The size of the picture in this review is not the same, the author should keep it the same size.
(2). In Scheme 7, the reaction conditions that lead from 20 to the formation of 21l and 21b should be "PTSA•H2O."
(3). In the reaction conditions in Scheme 11, the palladium catalyst used in the literature is Pd(OAc)2.
(4). The title for Scheme 12 should be "Photoredox/nickel dual catalysis en route to C–H vinylation."
(5). In the proposed mechanism in Scheme 17, during the transformation of intermediate B into intermediate C, the authors should include Zn to indicate its role as a reducing agent in this step.
(6). In the proposed mechanism in Scheme 18, the structure of intermediate C appears to be identical to that of intermediate B. The authors should carefully verify this.
(7). In line 179, it should be "Pd(0)" without the need for a superscript.
(8). In line 241, "product XX" should be corrected to "product 56.
(9). In the proposed mechanism in Scheme 17, the substrate is numbered as 72, and the author made an error by writing it as 71. This should be corrected.
(10). In line 241, " indole xxb" should be corrected to " indole 78.
(11). In Scheme 27, the author has depicted 1,2-dimethoxybenzene in the reaction conditions, but there is no corresponding description of it in the text.
(12). The inconsistency in the formatting of the references should be carefully reviewed and corrected by the author.
Author Response
Dear reviewer,
I would like to thank you for your efforts in reading, correcting and making recommendations to improve the manuscript. Please find below the answers to your questions and recommendations:
Comment (1). The size of the picture in this review is not the same, the author should keep it the same size.
Response: The size of the schemes has been unified
Comment (2). In Scheme 7, the reaction conditions that lead from 20 to the formation of 21l and 21b should be "PTSA•H2O."
Response: Corrected
Comment (3). In the reaction conditions in Scheme 11, the palladium catalyst used in the literature is Pd(OAc)2.
Response: Corrected
Comment (4). The title for Scheme 12 should be "Photoredox/nickel dual catalysis en route to C–H vinylation."
Response: The title has been modified.
Comment (5). In the proposed mechanism in Scheme 17, during the transformation of intermediate B into intermediate C, the authors should include Zn to indicate its role as a reducing agent in this step.
Response: Zinc has been incorporated into the scheme
Comment (6). In the proposed mechanism in Scheme 18, the structure of intermediate C appears to be identical to that of intermediate B. The authors should carefully verify this.
Response: Scheme 18 was modified, the structure of intermediate C was replaced with the structure of final product 51.
Comment (7). In line 179, it should be "Pd(0)" without the need for a superscript.
Response: Corrected
Comment (8). In line 241, "product XX" should be corrected to "product 56.
Response: Corrected
Comment (9). In the proposed mechanism in Scheme 17, the substrate is numbered as 72, and the author made an error by writing it as 71. This should be corrected.
Response: Corrected
Comment (10). In line 241, " indole xxb" should be corrected to " indole 78.
Response: Corrected
Comment (11). In Scheme 27, the author has depicted 1,2-dimethoxybenzene in the reaction conditions, but there is no corresponding description of it in the text.
Response: Comments on the role of 1,2-dimethoxybenzene during the preparation of papaverine have been added to the text.
Comment (12). The inconsistency in the formatting of the references should be carefully reviewed and corrected by the author.
Response: References have been checked. The found inconsistencies and their formatting have been corrected.
Sincerely,
Tomas Tobrman
Reviewer 2 Report
This review summarizes the applications of vinyl sulfonate and vinyl acetate as green alternatives for vinyl bromide in cross-coupling reactions. The preparation of vinyl sulfonates/acetate and their cross-coupling reactions are detailedly discussed. The author also points out that The reactivity of other electrophiles, particularly vinyl phosphates, is undeveloped, which severely limits the development of multicomponent and cyclization reactions using vinyl templates as a C2 synthon source. Overall, the review is well structured, analyzed, and discussed. I would like to recommend its publication after minor revisions.
1, Are there any synthetic applications reported of these couplings in the synthesis of drug and natural product molecules? To them more attention should be paid.
2, In the introduction part, a general mechanism is depicted in Scheme 1. However, the coupling reaction mechanism is highly dependent on the transition-metal catalyst. A radical process, or others, might also be involved. The different mechanism should be indicated when the detailed reaction is described. For example, Scheme 6, Scheme 8, Scheme 14, Scheme 16, Scheme 19, etc.
3, Page 26, Scheme 29, do the Hg(OAc)2-catalyzed couplings share the similar mechanism with the Ir-catalyzed couplings?
4, The author should take care of the concepts of cross-couplings and C-H activations. I suggest they should be separately discussed.
5, Scheme 30, compound 95 inside the cycle is wrongly drawn.
Minor editing of English language required.
Author Response
Dear reviewer,
I would like to thank you for your efforts in reading, correcting and making recommendations to improve the manuscript. Please find below the answers to your questions and recommendations:
Comment - 1, Are there any synthetic applications reported of these couplings in the synthesis of drug and natural product molecules? To them more attention should be paid.
Response: Selected applications of vinyl electrophiles have already been mentioned in the text, for example Scheme 27 or Scheme 7.
Comment - 2, In the introduction part, a general mechanism is depicted in Scheme 1. However, the coupling reaction mechanism is highly dependent on the transition-metal catalyst. A radical process, or others, might also be involved. The different mechanism should be indicated when the detailed reaction is described. For example, Scheme 6, Scheme 8, Scheme 14, Scheme 16, Scheme 19, etc.
Response: I consider the above comment to be beneficial. However, the problem I see is the enormity of the issue of cross-coupling reactions. If I decide to discuss the mechanisms of specific cross-coupling reactions, the clarity of the article, which is mainly focused on the use of alternative vinyl electrophiles in cross-coupling reactions, would be lost.
Comment - 3, Page 26, Scheme 29, do the Hg(OAc)2-catalyzed couplings share the similar mechanism with the Ir-catalyzed couplings?
Response: A note on the mechanism of mercury-catalyzed transvinylation has been added to the manuscript, including references.
Comment - 4, The author should take care of the concepts of cross-couplings and C-H activations. I suggest they should be separately discussed.
Response: The relation of C-H bond activation to the content of this review is mentioned in the introduction.
Comment - 5, Scheme 30, compound 95 inside the cycle is wrongly drawn.
Response: Compound 95 inside the catalytic cycle has been corrected.
Sincerely,
Tomas Tobrman